# Storage Stability of Atheroglitatide, an Echogenic Liposomal Formulation of Pioglitazone Targeted to Advanced Atheroma with a Fibrin-Binding Peptide

**DOI:** 10.3390/pharmaceutics15092288

**Published:** 2023-09-06

**Authors:** Melvin E. Klegerman, Tao Peng, Shao-Ling Huang, Brion Frierson, Melanie R. Moody, Hyunggun Kim, David D. McPherson

**Affiliations:** 1Division of Cardiovascular Medicine, Department of Internal Medicine, The University of Texas Health Science Center at Houston, Houston, TX 77030, USA; tao.peng@uth.tmc.edu (T.P.); shaoling.huang@uth.tmc.edu (S.-L.H.); brion.frierson@uth.tmc.edu (B.F.); melanie.r.moody@uth.tmc.edu (M.R.M.); david.d.mcpherson@uth.tmc.edu (D.D.M.); 2Department of Biomechatronic Engineering, Sungkyunkwan University, Suwon 16419, Republic of Korea

**Keywords:** liposome, storage stability, Arrhenius analysis, nanomedical formulation, ultrasound, molecular targeting, controlled release

## Abstract

We have conducted a stability study of a complex liposomal pharmaceutical product, Atheroglitatide (AGT), stored at three temperatures, 4, 24, and 37 °C, for up to six months. The six parameters measured were functions of liposomal integrity (size and number), drug payload (loading efficiency), targeting peptide integrity (conjugation efficiency and specific avidity), and echogenicity (ultrasound-dependent controlled drug release), which were considered most relevant to the product’s intended use. At 4 °C, liposome diameter trended upward, indicative of aggregation, while liposome number per mg lipid and echogenicity trended downward. At 24 °C, peptide conjugation efficiency (CE) and targeting efficiency (TE, specific avidity) trended downward. At 37 °C, CE and drug (pioglitazone) loading efficiency trended downward. At 4 °C, the intended storage temperature, echogenicity, and liposome size reached their practical tolerance limits at 6 months, fixing the product expiration at that point. Arrhenius analysis of targeting peptide CE and drug loading efficiency decay at the higher temperatures indicated complete stability of these characteristics at 4 °C. The results of this study underscore the storage stability challenges presented by complex nanopharmaceutical formulations.

## 1. Introduction

Thus far, 14 liposomal pharmaceutical products have been approved by the U.S. FDA for parenteral clinical use [1]. The active ingredients of these formulations include chemotherapeutic agents, antibiotics, antifungals, photosensitizers, analgesics, and vaccines. These are complex products consisting of multiple components that can independently degrade by varied mechanisms. Thus, stability assessment of liposomal products presents a challenge to pharmaceutical scientists, the initial step of which is to determine characteristics most important to the product’s intended use.

We have developed a novel product known as Atheroglitatide (AGT), a peptide-targeted echogenic liposome loaded with an anti-inflammatory drug (pioglitazone) for prevention of reatherogenesis after percutaneous intervention for peripheral arterial disease (PAD). Ultrasound (US) is applied during local injection of the product at the time of stent placement to facilitate liposome penetration into the arterial wall and trigger drug release at the site of the atherosclerotic plaque (atheroma) [2]. The echogenic liposomes (ELIP) reflect US due to lyophilization in the presence of D-mannitol, which allows ambient gas such as air to form nanobubbles within the liposomes. Besides the obvious diagnostic utility of a targeted US contrast agent, ELIP has advantages as an US-triggered controlled release drug carrier because of cavitation effects [3,4]. In this case, a lipid drug nanocarrier has been combined with US-triggered controlled release and molecular targeting to a disease marker [5,6,7,8].

This product can also be used to prevent reatherogenesis and restenosis after stent placement for percutaneous coronary intervention to restore blood flow in coronary arteries after acute myocardial infarction and for percutaneous intervention to restore blood flow after acute ischemic stroke or in cases of deep vein thrombosis. Pioglitazone (PGN) is a peroxisome proliferator-activated receptor-gamma (PPARγ) agonist that reduces pro-inflammatory cytokines [9,10] and generates anti-inflammatory and anti-arteriosclerotic effects [9,11,12]. The water-soluble form of the drug is marketed as a systemically administered antidiabetic agent [13].

The U.S. FDA Drug Stability Guidelines state that test conditions should be appropriate to the projected storage conditions of the product and specify the duration of storage for different proposed expiration dates, but allow accelerated testing for degradation of active pharmaceutical ingredients (APIs) at higher temperatures [14]. Examples of this approach involve the use of Arrhenius kinetics to determine the activation energy, *E_a_*, by measuring the degradation rate at several temperatures. Consequently, the degree of API degradation at a given temperature can be extrapolated to any time [15]. In addition, assessment of liposomal drug stability often involves measuring drug retention or release rate [16].

In this study, we measured six AGT properties pertinent to the product’s intended use at three temperatures for up to 6 months. We tested the ability to project extended product integrity at 4 °C, the intended storage temperature, using Arrhenius kinetics.

## 2. Materials and Methods

### 2.1. Preparation of Pioglitazone-Loaded Echogenic Liposomes (PGN-ELIP)

Echogenic liposomes (ELIP) comprised 1,2-distearoyl-sn-glycero-3-phosphocholine (DSPC), 1,2-dioleoyl-sn-glycero-3-phosphoethanolamine-N-[4-(p-maleimidophenyl)butyramide] (MPB-DOPE), 1,2-dioleoyl-sn-glycero-3-phosphocholine (DOPC), and cholesterol, all from Avanti Polar Lipids, Alabaster, AL, at a molar ratio of 52:8:30:10. The lipids (100 mg) and 10 mg pioglitazone (Cayman Chemical, Ann Arbor, MI, USA) were dissolved in 10 mL absolute ethanol (Fisher Scientific, Hampton, NH, USA). The mixture was heated to 90 °C to foster dissolution, loaded into a 10 mL glass syringe with a 27G needle, and injected into 110 mL of 0.2 µm-filtered, autoclaved water with stirring (1000 rpm). Spontaneous formation of unilamellar liposomes occurred with phase mixing. The liposomal dispersion was stirred for 30 min at room temperature. The final ethanol concentration in the liposomal dispersion was 14.8% *v*/*v*. The residual solvent was removed by rotary evaporation (RII, Buchi, Cornaredo, Italy). After removal of residual solvent, the liposomes were resuspended in 0.32 M mannitol to a concentration of 10 mg lipid/mL. We developed a more industrially scalable ethanol injection method for ELIP production rather than the classic chloroform dissolution-evaporation-rehydration method [17] and optimized it for drug loading capacity, liposomal size distribution, and echogenicity. ELIP were not extruded to produce uniform vesicles, since larger liposomes exhibited more advantageous acoustic properties and drug capacity, while a range of liposome sizes afforded versatility of ultra-sound-induced drug release characteristics over different frequencies.

### 2.2. Conjugation of a Fibrin-Binding Peptide (PAFb) to Pioglitazone-Loaded MPB-ELIP

The peptide, H-Gly-Pro-Arg-Pro-Pro-Gly-Gly-Gly-Cys-NH2 HCl (GPRPPGGGC), contains the pentapeptide shown to bind to fibrin, which is also known to be a marker for late-stage atheroma at the amino terminus [18,19,20,21]. The peptide was purchased via a custom source material agreement with Bachem Americas, Inc. (Torrance, CA, USA). Liposomal MPB-DOPE was conjugated to the carboxy-terminal cysteinyl thiol group via a thioether linkage after PGN loading by overnight reaction of 1–2 mg PAFb and 60–100 mg MPB-PGN-ELIP at pH 6.5–6.7 with stirring at room temperature. After the reaction, the peptide-conjugated liposomes were purified by centrifugation at 10,000 rpm for 10 min at room temperature. Pellets were washed twice centrifugally with 0.02 M phosphate-buffered saline, pH 7.4 (PBS) and resuspended in 0.32 M D-mannitol. The pooled suspension (10 mg lipid/mL) was distributed into 3 mL crimp-top vials (Wheaton, Sigma-Aldrich, St. Louis, MO, USA; 0.5 mL/vial), frozen for ≥2 h at −80 °C, and lyophilized (Labconco FreeZone 6-L, Kansas City, MO, USA) at a vacuum of <100 µ for 48 h. Vials containing lyophilized cakes (5 mg lipid/vial) were charged with octafluoropro-pane (OFP) at 1 atmosphere pressure prior to storage.

### 2.3. Evaluation of PAFb-Conjugated PGN-ELIP (Atheroglitatide, AGT)

We characterized AGT for echogenicity, PGN content, ELIP size and number, PAFb conjugation efficiency (CE), and targeting efficiency (TE), determined as specific avidity. For quantitation of echogenicity, three replicate images generated using a 4V1c-S transducer probe connected to a Siemens Acuson Sequoia 512 ultrasound system were captured with a frame grabber (EZ Grabber Diamond VC500, Diamond Multimedia, Chatsworth, CA, USA) situated between the PC and the ultrasound system operating at 2 MHz frequency and 0.2 MI. The mean gray scale values (MGSV; 0–255 scale) of captured images were determined with ImageJ software (Version 1.53, available online: https://imagej.nih.gov/ij/).

PGN, following dissolution of AGT in 80% ethanol, was measured using a Waters high performance liquid chromatography (HPLC) system fitted with a 6 mm diameter × 300 mm length YMC ODS-A 5 µm C18 column. The mobile phase was 60% acetonitrile/40% methanol; injection volume was 20 µL; detection wavelength was 269 nm; and flow rate was 1 mL per minute.

PAFb conjugation efficiency (CE) was measured using an inhibition enzyme-linked immunosorbent assay (ELISA) as previously described [22], based on a published protocol [23]. After blocking with 1% bovine serum albumin (BSA) in 0.05 M tris buffer with 0.02% sodium azide (conjugate buffer), biotinylated PAFb (20 µg/mL PBS-T) mixed with various concentrations of PAFb standard or AGT dilutions were incubated for 2 h at 37 °C, followed by washing and a one-hour incubation with streptavidin-conjugated alkaline phosphatase (Millipore Sigma; 3000× in conjugate buffer) at 37 °C. The net optical densities (A405) of biotinylated PAFb mixed with PAFb standards, including a 0-point (no PAFb), were used to construct an inhibition curve. PAFb in AGT was calculated from the exponential decay function of the standard curve. CE was expressed as µg PAFb/mg lipid and number of PAFb molecules per liposome, based on Multisizer enumeration. Specific avidity, as a measure of TE, is the product of CE expressed as number of peptide molecules per liposome and fibrin binding affinity expressed as association constant in M-1 and was determined as previously described [23].

Size distribution analysis of AGT by dynamic light scattering (DLS) using a Malvern Zetasizer instrument (Malvern Paralytical Ltd., Malvern, UK) confirmed that liposomes < 0.4 µm in diameter were not recovered during centrifugal purification following PAFb conjugation. Therefore, particle size characterization, as well as ELIP enumeration, was performed only with the Beckman Coulter Multisizer 4 instrument (Beckman Coulter, Brea, CA, USA), using a 20 µm aperture tube, which measures particles between 0.4 and 10.0 µm.

### 2.4. Stability Protocols

Thirty-five aliquots of lyophilized AGT in crimpled glass vials were stored at each temperature (4, 24, and 37 °C) in the dark. At each time point (0, 1, 2, 3, 4, 5, 6 months at 4 °C; 0, 1, 2, 3, 4 weeks, 2, 3 months at 24 °C; 0, 3 days, 1, 2, 4, 5, 8 weeks at 37 °C), 5 vials were reconstituted with 0.5 mL nanopure water each for replicate analyses. The temperatures selected included the intended storage temperature of 4 °C and an elevated temperature (37 °C) at which product degradation is expected to occur. Ambient temperature (24 °C) was added, since a minimum of three temperatures is required for an Arrhenius analysis.

The days selected for sampling at each temperature were an estimate of the period over which product degradation was expected to occur, with the 6-month period for 4 °C being the minimum recommended by FDA guidelines [14]. Regarding the timing of analyses, echogenicity was determined immediately after AGT reconstitution on the day listed. All other analyses were performed on the reconstituted product the following day after storage at 4 °C.

### 2.5. Arrhenius Calculations

The Arrhenius equation is as follows:*k* = *A*exp(−*E_a_*/*RT*)(1)
where *k* is the rate constant of a chemical reaction, *A* is the initial concentration of reactant, *E_a_* is the activation energy of the reaction, *R* is the universal gas constant (1.987 cal/mole for calculation of *E_a_* in kcal/mole), and *T* is the temperature in degrees Kelvin. Converting to natural logs yields:ln*k* = −*E_a_*/*RT* + ln*A*(2)

Thus, plotting ln*k* vs. 1/*T* produces a regression line of slope −*E_a_*/*R* and *y*-intercept = ln*A*. This is known as the Arrhenius plot. If degradation rates governed by a discrete chemical reaction are measured at a minimum of three temperatures, *E_a_* of the reaction can be derived and the Arrhenius equation can be used to calculate k at any temperature and the extent of reactant degradation at any time [15], which is the procedure utilized in this study. All downward trending rates for CE and PGN loading efficiency were approximated as first-order reactions by subjecting them to linear regression analyses [15].

### 2.6. Statistics

Replicate values for each parameter at each time point (n = 5) were subjected to normal variance analysis. Differences between populations at successive time points and the population at 0-time were assessed by Student’s *t*-test; *p* ≤ 0.05 was considered to represent a significant difference. Linear and non-linear regressions with variance data were performed using SigmaPlot (Inpixon, Palo Alto, CA, USA).

## 3. Results

We developed stability parameters for the AGT product and methods to assess them over several previous studies [2,17,22].

The echogenicity quantitation for the 4 °C 0-point samples is shown in Figure 1. MGSV vs. AGT lipid concentration corresponds to a hyperbolic relationship. For the echogenicity index of the product, we chose the maximum MGSV with specification of the concentration.

The number-average distribution of a 0-point AGT sample, measured with a Beckman-Coulter Multisizer 4 instrument (Beckman Coulter Life Sciences, Indianapolis, IN, USA), is shown in Figure 2. The median value is used for quality control purposes. A 100 µL volume sample is counted, based on the Coulter principle, which involves perturbation of impedence produced by particles drawn through an aperture by a manometric vacuum. The advantage of the Multisizer is that it enables simultaneous particle enumeration and size distribution analysis, in this case down to 400 nm utilizing a 20 µm aperture tube. Analysis of the same sample with a dynamic light scattering instrument, such as the Malvern Zetasizer, enables enumeration of particles smaller than 400 nm. We had already determined that our centrifugal purification method for AGT eliminated all liposomes smaller than 400 nm.

An HPLC profile of a 0-point AGT sample is shown in Figure 3. The PGN peak is identified at 6.576 min rt.

PGN standard curves comprising PGN alone or PGN in the presence of 20-fold dilute plain ELIP dispersed in 80% ethanol were linear from 3 to 100 µg/mL (Figure 4). Y-intercepts were not significantly different from zero. AGT 0-point sample dilutions (25× and 50×) clustered around the curves, indicating that they obeyed the same linear relationship as the standards.

A log-log transform of a PAFb ELISA inhibition curve with superimposition of 16 sample points from 4 °C study is shown in Figure 5.

We produced 10 lots of AGT over a period of 14 months, including the three lots used for the stability study reported here. Mean and variance data for the six stability characteristics of these lots are tabulated in Table 1.

The mean values (with variances, n = 3–5) for six major Atheroglitatide characteristics at each time point and temperature are listed in Table 2. At 4 °C, echogenicity trended downward, becoming significantly different from the 0-point value at three months, while liposome diameter trended upward, indicative of aggregation, becoming significant at two months. The PGN loading efficiency nearly doubled at four months and remained elevated with no trend thereafter. At 24 °C, liposome diameter trended downward, becoming significant at one week, while PGN loading efficiency dropped by one-third at one week, remaining stable thereafter. At 37 °C, echogenicity initially trended upward, reaching a plateau at one week and decreasing below the 0-point value at eight weeks. PGN loading efficiency trended downward, becoming significant at three days. Conjugation efficiency also trended downward, becoming significantly lower than the 0-point value at eight weeks. Despite this, the targeting efficiency (TE) increased significantly at eight weeks, mainly due to a higher peptide binding affinity measured at that time.

Major trending parameters at each temperature are shown in Figure 6 as percent of 0-point value. In addition to echogenicity and liposome diameter at 4 °C, which attained significance, liposome number per mg lipid also trended downward, while not becoming significantly different from the 0-point value. At 24 °C, while liposome diameter and PGN loading efficiency became significantly different from the 0-point value, only conjugation efficiency and targeting efficiency, partially derived from CE, showed a progressive downward trend, indicative of a deterioration of peptide integrity. At 37 °C, both conjugation efficiency and PGN loading efficiency showed a progressive downward trend, the latter indicative of API degradation.

Arrhenius analysis of CE and PGN loading efficiency data in degradation rates of µM PAFb/d and mM PGN/d yielded activation energies of 36.6 and 48.3 kcal/mole, respectively. The former is very close to the range of values published for thioether degradation [24], while the latter is consistent with *E_a_* values found for degradation of aromatic ether linkages [25], implying that API degradation proceeds through scission of the ether linkage with release of a pyridine derivative. Use of the Arrhenius equation to calculate the degradation rate at 4 °C in both cases yielded values of −1.80 × 10^−5^ µg PAFb/mg lipid/d and −2.20 × 10^−6^ µg PGN/mg lipid/d. At the preferred storage temperature, loss of the conjugated peptide and encapsulated drug will be negligible.

## 4. Discussion

Nanomedical formulations, particularly those involving liposomal drug carriers, present special challenges regarding assessment of storage stability. Although pharmaceutical formulations are often more complex than the API alone, involving excipients and macrocarriers such as capsules or soluble stabilizers for liquids, nanomedical formulations include complex configurations of matter on the spatial scale of the API, as well as those found in more conventional pharmaceutical entities. Degradation of the API usually occurs through discrete chemical reactions and can be assessed by chemical kinetic analysis, but other mechanisms of nanopharmaceutical degradation may be more difficult to define. In the case of liposomal formulations, liposomal integrity, being dependent on phospholipid/lipid structure, drug-liposome interactions, and colloidal physicochemical dynamics, is an important consideration. Chemical alterations of nanopharmaceutical components may generate toxicities or reduce product effectiveness. It may be advisable to focus stability assessments on formulation aspects that are most relevant to the intended use of the product, at least initially.

In the case of our product, Atheroglitatide, relevant aspects are the potency of the anti-inflammatory drug, pioglitazone, the integrity of the fibrin-targeting peptide, and ELIP echogenicity (which controls drug release characteristics) and physical integrity. The first two aspects are likely to be governed by features that can be degraded through discrete chemical reactions, while the latter two may be more difficult to define and, therefore, are best determined empirically at the intended storage temperature. In the latter case, reciprocal trends were observed between particle diameter and number per unit volume, which would be expected for liposome aggregation. The PAFb peptide is relatively hydrophobic, increasing the probability of aggregation, but this effect was not seen at higher temperatures, contrary to expectations [26]. The upper limit of desirable particle size was achieved after six months of storage at 4 °C, thus defining the product expiration under those conditions.

Additionally, a downward trend in AGT echogenicity, which also reached a lower limit of tolerance at six months, was observed at 4 °C. In fact, a progressive increase in echogenicity trend rates, from negative to positive, with increasing temperature was seen and could be related to decreasing gas solubility with increasing temperature, resulting in enhanced bubble formation. However, an upper plateau of echogenicity was reached after one week at 37 °C, defining the OFP gas saturation point.

Accelerated aging at increased temperatures with Arrhenius analysis proved to be a valuable strategy for assessment of stability at the intended storage temperature for the two product attributes subject to degradation by discrete chemical reactions, and implicated the likely degradation mechanisms. Based on calculated activation energies, it is likely that the PAFb peptide can be lost by cleavage of the thioether linkage between the peptide and the MPB-PE maleimide, while the effective inactivation of pioglitazone that would result in decreased drug concentrations detected by HPLC would require the loss of pyridine from the drug. UV absorbance maxima of free pyridine in aqueous solution occur between 250 and 260 nm, with virtually no absorbance at 269 nm [27], which is the detection wavelength of the HPLC protocol. However, Arrhenius analysis indicated that these reactions are negligible at 4 °C.

We believe that the comprehensive nature of this study, involving multiple aspects of a nanoparticulate formulation, is a novel application of an established pharmaceutical procedure that will become more common as nanomedical therapeutics enter the marketplace. The results of this study support the expectation that a potential drawback of complex nanopharmaceutical formulations is a shortening of the effective storage duration due to a multiplication of potential degradation mechanisms. This expectation should be factored into gauging the feasibility of nanopharmaceutical products.

The findings of this study also have implications for the stability of the product in vivo. We previously showed that the AGT product was effectively delivered to the por-cine arterial wall after balloon injury of the endothelium [2]. The AGT product was administered intra-arterially to denuded iliofemoral and carotid segments through an US-emitting EKOSTM catheter in Yucatan miniswine. For assessment of product delivery, AGT components were labelled with stable adjuncts so that the IHC staining procedures could be done at the time of tissue processing, preventing degradation of the formulation or the labels with time. We confirmed PGN delivery into the arterial wall by bioanalytical quantitation of the PGN associated with the artery and demonstrated that ultrasound promotes penetration of the targeted ELIP deeper into the vessel wall beyond the intima, thus facilitating therapeutic delivery throughout the arterial wall and potentially promoting more complete inhibition of neo-atheroma formation. We also found evidence by IHC that PGN migrated from the ELIP within the medial and adventitial layers of the arterial wall.

The expectation of product efficacy was supported by an earlier study in which an analogous product was targeted to atheroma by conjugation of PGN-loaded ELIP to a monoclonal antibody specific for intercellular adhesion molecule-1 (ICAM-1) [17]. Therapeutic delivery to the superficial/intima and deeper/media layers of the arterial wall using our strategy of sequential delivery of a nitric oxide donor (NO-ELIP or nitroglycerin) and a targeted PGN-ELIP formulation into the stented arteries in LDL-receptor deficient miniswine using the EKOS catheter resulted in attenuation of neointimal growth. This stability study showed that two-thirds of the drug remains intact after two months at physiologic temperature. Thus, it is likely that PGN deposited in atheroma by this approach will continue to exert anti-inflammatory and anti-atherosclerotic effects for extended periods of time. This hypothesis will be tested in a future pharmacodynamic study.

## 5. Conclusions

A storage stability study of a complex nanopharmaceutical formulation has demonstrated that both temperature-specific and accelerated Arrhenius-based assessments can be used to determine the relative contributions of multiple formulation components to comprehensive nanomedical product stability. The findings for Atheroglitatide indicate probable feasibility for commercial development of a product which is likely to exhibit novel pharmacodynamic characteristics.

## Figures and Tables

**Figure 1 pharmaceutics-15-02288-f001:**
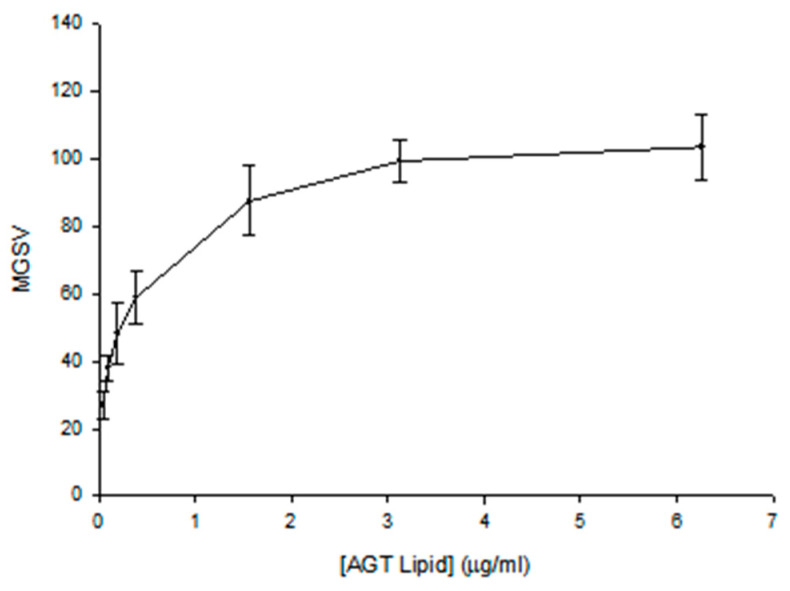
Dose response of AGT echogenicity, converted by videodensitometric analysis to mean gray scale values (MGSV) using a 0 to 255 scale of gray shades.

**Figure 2 pharmaceutics-15-02288-f002:**
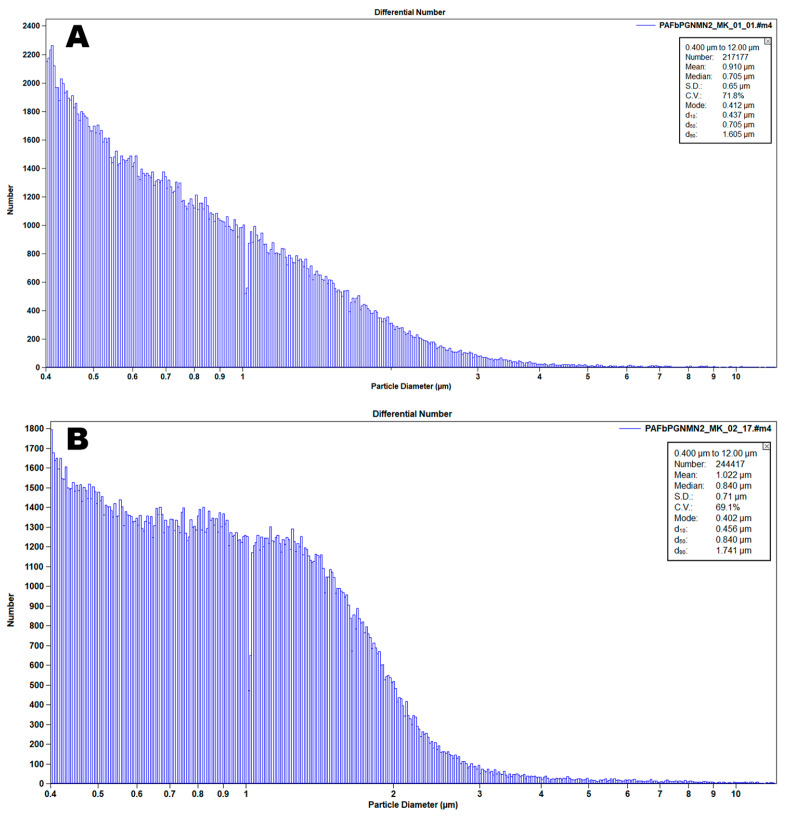
Multisizer 4 number-average particle size (spherical diameter) distribution of (**A**) 0-point and (**B**) 6-month AGT samples, 8000× dilution, 100 µL.

**Figure 3 pharmaceutics-15-02288-f003:**
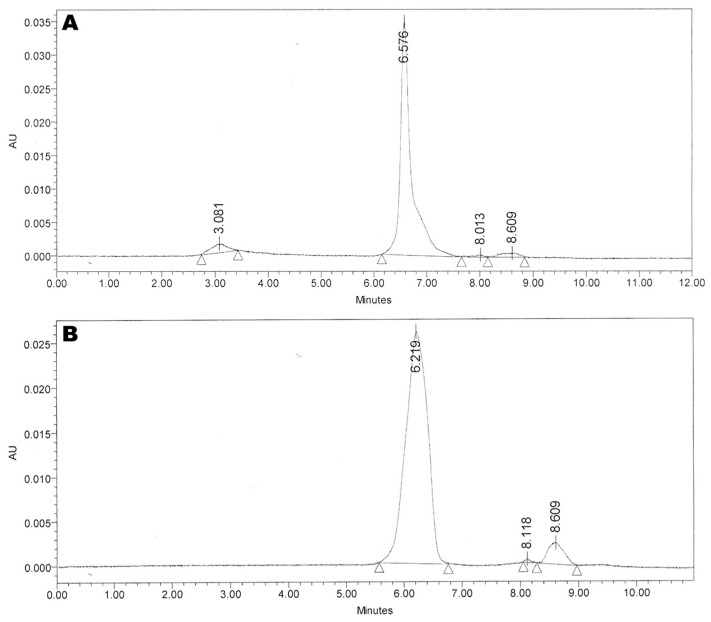
HPLC profiles of (**A**) 0-point and (**B**) 6 months AGT 4 °C stability samples. Triangles in the profiles denote points of automatic integration.

**Figure 4 pharmaceutics-15-02288-f004:**
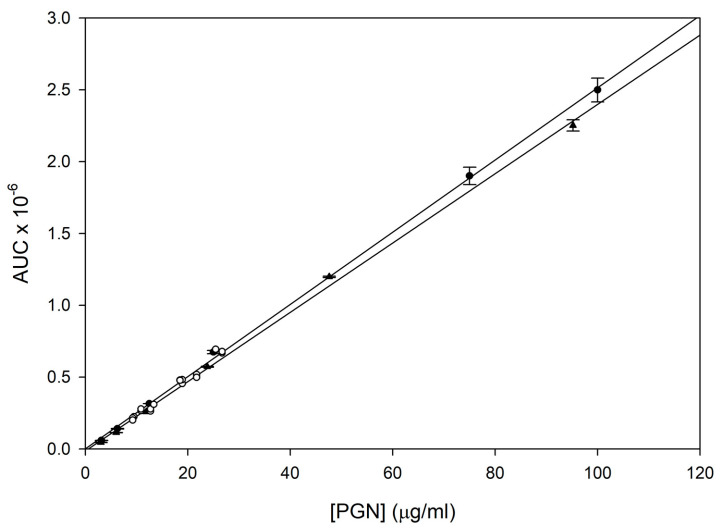
HPLC composite standard curves for PGN (solid circles; y = 0.0251x + 0.0011) and PGN in 20× dilute ELIP (unconjugated, unloaded; solid triangles; y = 0.0241x − 0.0143) dispersed in 80% ethanol. Points are means of 3 determinations; bars = SD. AGT 0-time 4 °C stability sample points (open circles) are superimposed on the curves.

**Figure 5 pharmaceutics-15-02288-f005:**
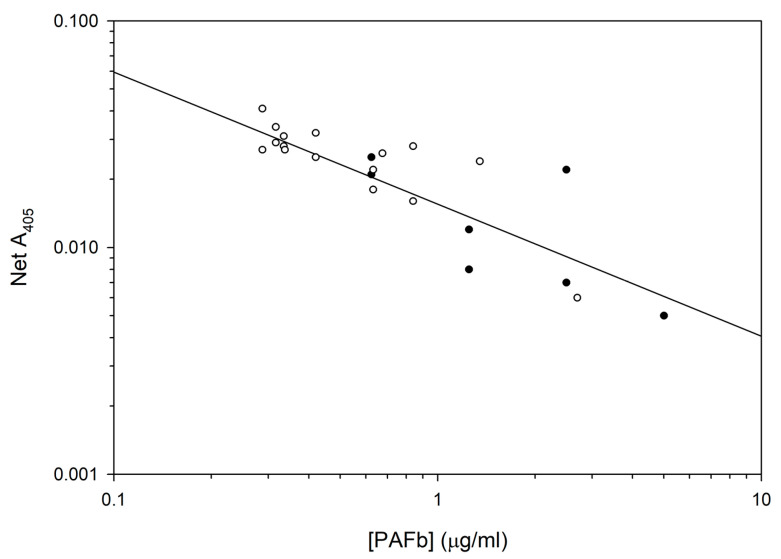
PAFb inhibition ELISA standard curve (solid circles) with 16 sample points from 4 °C study, showing superimposibility, indicating identity of ELIP-conjugated PAFb with the PAFb standard. Log y = −0.583 log x − 1.809; r = 0.705.

**Figure 6 pharmaceutics-15-02288-f006:**
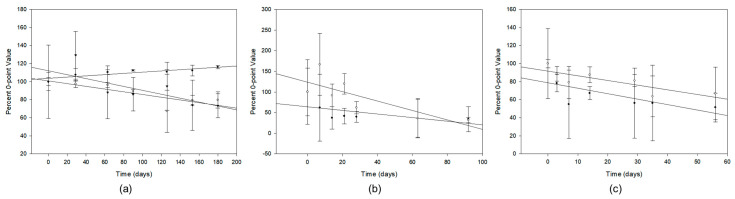
Trending parameter regressions. (**a**) 4 °C data. Closed circles, liposome diameter; open circles, echogenicity; inverted triangles, particle number. (**b**) 24 °C data. Closed circles, conjugation efficiency; open circles, targeting efficiency (specific avidity). (**c**) 37 °C data. Closed circles, conjugation efficiency; open circles, PGN loading efficiency. Data points are mean percent 0-point values ± SD, n = 4–5 as specified in Table 1.

**Table 1 pharmaceutics-15-02288-t001:** Major characteristics of ten lots of Atheroglitatide. Mean ± SD.

Echogenicity (MGSV @2 MHz)	Liposome Diameter (nm)	Liposome No. (mg Lipid^−1^ × 10^−9^)	Pioglitazone (µg/mg Lipid)	CE (µg/mg Lipid)	Specific Avidity (M^−1^/Lip. × 10^−12^)
103.1 ± 6.9	746.8 ± 98.3	3.24 ± 1.19	78.2 ± 32.8	2.01 ± 1.31	0.114 ± 0.061

**Table 2 pharmaceutics-15-02288-t002:** Major characteristics of Atheroglitatide upon storage at different temperatures. Mean ± SD (n = 5, unless otherwise specified).

Temp. (°C)	Time (Days)	Echogenicity (MGSV @2 MHz)	Liposome Diameter (nm)	Liposome No. (mg Lipid^−1^ × 10^−9^)	Pioglitazone (µg/mg Lipid)	CE (µg/mg Lipid)	Specific Avidity (M^−1^/Lip. × 10^−12^)
4	0	103.1 ± 10.2	704.2 ± 31.3	2.57 ± 1.04	55.7 ± 9.3	0.42 ± 0.13	0.081 ± 0.031
	29	100.4 ± 3.9	756.9 ± 52.6	3.32 ± 0.88	58.5 ± 11.6	0.27 ± 0.39	0.036 ± 0.050
	63	99.3 ± 2.7	778.2 ± 19.9 *	2.26 ± 0.66	50.4 ± 9.7	0.99 ± 0.32 *	0.063 ± 0.032
	90	91.9 ± 1.8 *	791.5 ± 6.6 *	2.21 ± 0.41	47.2 ± 9.1	0.79 ± 0.34	0.046 ± 0.018
	126	69.4 ± 16.2 *	781.2 ± 20.0 *	2.44 ± 0.65	81.4 ± 12.5 *	0.43 ± 0.52	0.049 ± 0.045 †
	153 †	82.0 ± 4.3 *	790.5 ± 46.2 *	1.90 ± 0.53	87.4 ± 11.2 *	0.43 ± 0.38	0.230 ± 0.119 ‡
	180	82.1 ± 7.3 *^,^†	817.6 ± 15.5 *	1.88 ± 0.25	90.9 ± 18.2 *	1.11 ± 1.91	0.897 ± 1.264 †
24	0	91.8 ± 11.4	710.0 ± 28.0	2.82 ± 0.93	135.0 ± 10.1	2.46 ± 1.42 †	0.138 ± 0.108 †
	7	84.7 ± 10.4	668.4 ± 8.2 *	1.89 ± 0.13	90.1 ± 9.6 *	1.52 ± 1.23	0.230 ± 0.173
	14	90.8 ± 5.6	671.6 ± 16.2 *	2.01 ± 0.16	89.5 ± 6.2 *	0.91 ± 0.25	0.126 ± 0.035
	21	86.3 ± 5.9	662.2 ± 13.3 *	1.99 ± 0.16	88.6 ± 8.1 *	1.02 ± 0.19	0.165 ± 0.041
	28	92.3 ± 6.9	674.2 ± 11.3 *	1.81 ± 0.06	87.9 ± 11.4 *	0.98 ± 0.13	0.085 ± 0.013
	63	95.6 ± 3.1	651.0 ± 19.9 *	2.00 ± 0.14	56.3 ± 28.4 *	0.89 ± 0.42	0.050 ± 0.023
	92	95.7 ± 3.1	655.8 ± 15.5 *^,^†	2.03 ± 0.41 †	90.9 ± 26.4 *	0.84 ± 0.25	0.044 ± 0.003 †
37	0	101.9 ± 10.5	685.0 ± 18.3	2.02 ± 0.13	69.3 ± 3.2	1.18 ± 0.46	0.061 ± 0.025
	3	103.7 ± 4.2	657.4 ± 16.7 *	2.10 ± 0.46	61.4 ± 5.1 *	0.93 ± 0.09	0.036 ± 0.006
	7	105.7 ± 4.3	670.4 ± 9.2	1.99 ± 0.05	54.8 ± 9.8 *	0.65 ± 0.25	0.047± 0.019
	14	104.1 ± 1.5	681.6 ± 20.5	1.94 ± 0.10	60.7 ± 5.4 *	0.80 ± 0.06	0.083 ± 0.010
	29	105.6 ± 4.7	678.4 ± 5.1	1.98 ± 0.10	56.2 ± 3.7 *	0.67 ± 0.26	0.077± 0.031
	35	105.7 ± 3.7	672.2 ± 8.4	2.04 ± 0.08 †	44.1 ± 10.0 *	0.67 ± 0.28	0.094 ± 0.046 †
	56	96.5 ± 3.5	645.8 ± 14.9 *	1.96 ± 0.14	46.3 ± 13.5 *	0.61 ± 0.07 *	0.175 ± 0.028 *

* *p* < 0.05 vs. 0-point value. † n = 4, ‡ n = 3.

## Data Availability

Original data may be obtained on request from Melvin E. Klegerman.

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
