# Peer review of "Storage Stability of Atheroglitatide, an Echogenic Liposomal Formulation of Pioglitazone Targeted to Advanced Atheroma with a Fibrin-Binding Peptide"

_pharmaceutics, 2023, doi:10.3390/pharmaceutics15092288_

Round 1
Reviewer 1 Report (Previous Reviewer 1)
I appreciate the answers to my questions and the improvements made to the document based on the suggestions.
Therefore, I consider that the manuscript has improved.
Author Response
Please see the attachment.

Reviewer 2 Report (New Reviewer)
Authors studied the stability of Atheroglitatide, a complex liposomal product at 4, 24, and 37°C. The liposome integrity, drug payload, peptide integrity, specific avidity, and ultrasound-based drug release were measured. This study is novel and essential to perform studies on various physical parameters during the course of storage. This manuscript is of high significance for those involved in liposomal research field and recommend for publication. Below comments are very minor.
¾ Results section, line 191-192 Please modify “quality control parameters” to “stability parameters.”
¾ Please include a chromatogram of a stability sample (after a certain time) along with 0-point sample
¾ What is the projected storage stability shelf-life of this project beyond 6 months?
Author Response
Please see the attachment.

Reviewer 3 Report (New Reviewer)
The authors describe the shelf-life and storage stability of AGT formulation at different temperatures via a thorough study of various formulation parameters and echogenic release. The manuscript is scientifically sound and well-written (although believed to have undergone an initial stage of review) and can be accepted in the present form for publication.
Author Response
Please see the attachment.

This manuscript is a resubmission of an earlier submission. The following is a list of the peer review reports and author responses from that submission.
Round 1
Reviewer 1 Report
First of all, congratulations for the work done. I think that some sections need to be further developed, such as collecting comparative results of all temperatures, figures, etc. and in the discussion section some more bibliographical reference should be included and expanded.
I would like to resolve some questions and make some comments. Thanks in advance.
- I reiterate that in the results section, data or comparative representations of the other two temperatures will probably appear in all the studies carried out, although the conclusions will confirm that the best results obtained correspond to the stability of the liposomes at 4 degrees.
- On line 163, correcting the symbol for degrees Celsius.
- The paragraph (line 203-205) is not very well understood. From those 10 lots, 3 of them were used to do the stability studies, right?
- A high variability in the conjugation efficiency (CE) of PAFb 99 is observed. Do you think that increasing the number of samples could decrease the standard deviation?
- Was the polydispersity index of the size of the liposomes determined? Which was? Was the size of liposomes homogenized?
- What criteria was taken into account to select the days at each temperature to perform the different echogenicity tests, liposome diameter... Was it done randomly?
- In Table 2, in the Specific Avidity parameter at 180 days, a deviation above the average is observed. Is it an error or should the number of samples be increased?
- I think it would be necessary to include some more reference. The section of the discussion should be further elaborated.
Reviewer 2 Report
The articles entitled “Storage Stability of Atheroglitatide, an Echogenic Liposomal 2 Formulation of Pioglitazone Targeted to Advanced Atheroma 3 with a Fibrin-binding Peptide” describes the storage stability and few quality attributes of developed liposomal formulation. Authors didn’t mention about formulation optimization, failed to characterize complete critical quality attributes of the formulation. The manuscript needs improvements in terms of; the results section should be elaborated with the references from the previous literature. The manuscript in the current is not up to the journal standard.
1. Please justify the selection of storage temperatures 4, 24, and 37°C.
2. Authors should highlight the novelty of this research work in the introduction section. Also, quote If there are any similar works available in the literature.
3. Page 3, line 70, Was there any rationale for choosing/fixing the lipid concentrations at a molar ratio of 52:8:30:10?
4. Mention the experimental conditions and details on lyophilization of liposomal formulation
Reviewer 3 Report
The authors have not shown any potential application(s) of the said formulation. Also, the details about the peptide used is missing - whether it was an outsourced synthetic peptide or was it synthesized in the lab?
By looking at the current level of the journal and its readership, I will not recommend its publication in Pharmaceutics just based on the stability data of a formulation. Authors should show and discuss its actual applications and compare it with the previous reported formulations or API alone.
The overall quality is fine.
